# Joint Power and Time Allocation in Hybrid NoMA/OMA IoT Networks for Two-Way Communications

**DOI:** 10.3390/e24121756

**Published:** 2022-11-30

**Authors:** Dong-Hua Chen, En-Hua Jiang

**Affiliations:** School of Physics and Electronic Information, Huaibei Normal University, No. 100 Dongshan Road, Huaibei 235000, China

**Keywords:** non-orthogonal multiple access, time division multiple access, internet of things, convex optimization

## Abstract

This article investigates two-way communications between an access point (AP) and multiple terminals in low-cost Internet of Things (IoT) networks. The main issues considered are the asymmetric transmission traffic on the uplink (UL) and downlink (DL), and the unbalanced receivers processing capability at the AP and the terminals. As a solution, a hybrid non-orthogonal multiple access/orthogonal multiple access (NoMA/OMA) scheme together with a joint power and time allocation method is proposed to address these issues. For the system design, we formulated the optimization problem with the aim of minimizing the system power and satisfying the UL and DL transmission rate constraints. Due to the coupling of power and time variables in the objective function and the multi-user interference (MUI) in the UL transmission rate constraints, the formulated problem is shown to be non-linear and non-convex and thus is hard to solve. To obtain a numerical, efficient solution, the original problem is first reformulated to be a convex one relying on the successive convex approximation (SCA) method, and then a numerical efficient solution is thus obtained by using an iterative routine. The proposed transmission scheme is shown to be not only physically feasible but also power-efficient.

## 1. Introduction

Non-orthogonal multiple access (NoMA) allows simultaneous transmission for multiple users on the same frequency via power domain superimposed coding, thereby achieving improved spectrum efficiency as compared to conventional orthogonal multiple access (OMA), such as time division multiple access (TDMA) [1,2]. Since its advent, NoMA has appeared in a variety of communication scenarios from Sub-6 GHz communications, ground networks, and cognitive radio networks, to mmWave and terahertz communications, space–air networks, and energy harvesting communication networks [3,4,5]. Although the investigation on 6G is still in its infancy, NoMA, in addition to multiple input multiple output (MIMO), has been recognized as a potential enabling technique for the future 6G networks, and therefore the two techniques of NOMA and MIMO will work jointly in the 6G networks [6,7].

As a branch of NoMA-based communication networks, NoMA-based wireless access networks are the most widely used. Till now, most study on the NoMA based wireless access networks has been focused on one-way transmissions wherein the uplink (UL) and downlink (DL) are separated considered [8]. For separated design on the UL and DL, it is difficult to make full use of available transmission resources to fulfill the network-wide performance optimum. On this account, some works studied relay aided two-way transmissions for wireless powered relaying networks and NoMA-based wireless networks [9,10,11,12]. From the performance analysis perspective, one investigation [9] studied relay aided two-way information exchange between two groups of users. Within the cooperative NoMA framework and from the optimization perspective, the authors of [10] investigated two-way communications between a pair of users, and the authors of [11] researched two-way communications between a base station and several cellular edge users.

For two-way communications, the above-mentioned works considered equal-length UL and DL transmission time, and for multiuser access, both the UL and DL used NoMA; and for system design, transmit power is the only resource that can be optimized. In some low-cost Internet of Things (IoT) networks, traffic levels on the UL and DL are asymmetric, and the traffic on one link may be heavier than on the other [13]. For this case, a time division duplex (TDD) with equal-length transmission duration cannot adapt to the asymmetric traffic. On the other hand, NoMA receiver involves complex multiuser detection (MUD), and it is a challenge for low-cost IoT terminals/devices (IoDs) to cope with the complex MUD, and thus, there may be difficulties for the DL to adopt NoMA in the low-cost IoT applications [14].

The computational complexity for the receiver performing MUD increases exponentially with the number of NoMA users, and thus the cost of the NoMA performance improvement is the soaring receiver complexity. For performance and complexity compromise, some works resort to hybrid NoMA/OMA for reduced NoMA size, in which users are organized into multiple groups so that NoMA is used in each group, and OMA is applied across different groups [15,16,17,18,19]. We specifically highlight that these studies on hybrid NoMA/OMA only considered one-way transmissions.

For two-way communications in low-cost IoT applications, this article proposes a scalable TDD protocol that combines NoMA and TDMA. In the UL, spectrum-efficient NoMA is used relying on the access point (AP)’s powerful processing capability, and in the DL, TDMA is adopted to facilitate signal detection of the IoDs with weak processing capability. For the system design, we jointly optimized the transmission time on the UL and DL, the transmission time for the DL TDMA slots, and the transmit power for the IoDs in both UL and DL directions. The proposed two-way communications with scalable TDD and hybrid NoMA/OMA are distinguished from both the equal-length UL and DL two-way transmission schemes and the one-way hybrid NoMA/OMA schemes mentioned above, and can well address the asymmetric quality of services (QoS) requirement and the unbalanced receivers processing capability between UL and DL.

## 2. System Model

We consider two-way communications between an AP and *K* IoDs. All the nodes are equipped with single antenna and work in half duplex mode. The IoDs are assumed to have limited processing capability, and the traffic on the UL and DL are asymmetric. To adapt to the transmission scenarios, we propose a scalable TDD transmission scheme by combing NoMA and TDMA. As the associated transmission protocol in Figure 1 showed, the transmission time is divided into two proportions of τ and 1−τ for the UL and DL, respectively. On the UL, NoMA is applied by exploiting the AP’s strong processing capability, and on the DL, the IoDs with limited processing capability cannot afford MUD for NoMA, and TDMA is thus adopted. Furthermore, the transmission time for TDD, and that for TDMA, is scalable and can be adjusted according to different traffic requirements on the UL and DL.

On the UL, the received signal at the AP is
(1)$$y_{\mathrm{AP}}=\sum_{k=1}^{K}\sqrt{p_{U,k}}h_{k}x_{U,k}+n_{\mathrm{AP}},$$
where $p_{U,k}$, $x_{U,k}$, and $h_{k}$ are the transmit power, the energy-normalized transmit data, and the channel response from the IoD *k* to the AP, respectively. $n_{\mathrm{AP}}∼CN\left(0,\sigma_{\mathrm{AP}}^{2}\right)$ is the associated additive white Gaussian noise (AWGN), where $CN\left(0,\sigma^{2}\right)$ denotes a circularly symmetric, complex Gaussian distribution with zero mean and covariance $\sigma^{2}$.

On the DL, the received signal at the IoD *k* is
(2)$$y_{k}=\sqrt{p_{D,k}}h_{k}x_{D,k}+n_{k},$$
where $p_{D,k}$ and $x_{D,k}$ are the transmit power and the energy-normalized transmit data from the AP to the IoD *k*, respectively. $n_{k}∼CN\left(0,\sigma_{k}^{2}\right)$ is the AWGN. Due to TDMA, the multiuser interference is absent at the IoD receivers. Meanwhile, due to TDD, the channel reciprocity was exploited when expressing the UL and DL received signals.

According to the transmission protocol discussed in Figure 1, the UL and DL transmission rates for the IoD *k* are
(3)$$R_{U,k}=\tau \log_{2}\left(1+\frac{{\left|h_{k}\right|}^{2}p_{U,k}}{\sum_{j=k+1}^{K}{\left|h_{j}\right|}^{2}p_{U,j}+\sigma_{\mathrm{AP}}^{2}}\right),$$
(4)$$R_{D,k}=\tau_{k}\log_{2}\left(1+\frac{{\left|h_{k}\right|}^{2}p_{D,k}}{\sigma_{k}^{2}}\right),$$
respectively, where $\tau_{k}$ is the DL transmission duration for the IoD *k* that satisfies $\sum_{k=1}^{K}\tau_{k}=1-\tau$. Note that the optimal decoding order for NoMA follows the rules of descending channel gains [20], and we have assumed ${\left|h_{1}\right|}^{2}\geq {\left|h_{2}\right|}^{2}\geq \cdots \geq {\left|h_{K}\right|}^{2}$ when formulating the UL transmission rate.

For the system design, we formulate the optimization problem with the aim of minimizing the system power and satisfying the UL and DL transmission rate constraints. According to the transmission protocol described previously, the system parameters $\left\{\tau ,\tau_{k},p_{D,k},p_{U,k}\right\}$ must be available for the system to work properly, and as a matter of course, these parameters become the optimization variables. The formulated problem is given by
(5)$$\begin{array}{ccc} \min_{\left\{\tau ,\tau_{k},p_{D,k},p_{U,k}\right\}} & & \tau \sum_{k=1}^{K}p_{U,k}+\sum_{k=1}^{K}\tau_{k}p_{D,k}\\ s.t. & C1: & R_{U,k}\geq {\bar{R}}_{U,k},\\ & C2: & R_{D,k}\geq {\bar{R}}_{D,k},\\ & C3: & \sum_{k=1}^{K}\tau_{k}=1-\tau ,\\ & C4: & 1\geq \tau \geq 0,\\ & C5: & \tau_{k}\geq 0,\\ & & k=1,2,\cdots ,K, \end{array}$$
where the constraints C1 and C2 guarantee the minimal UL and DL transmission rates are not below the thresholds ${\bar{R}}_{U,k}$ and ${\bar{R}}_{D,k}$, respectively. The constraints C3, C4, and C5 make the transmission time not only conform to the rules of the TDD and TDMA but also physically feasible. Note that the system power expressed by the objective function consists of two parts, with the former and the latter being contributions from the UL and DL transmissions, respectively. Due to the coupling of power and time variables in the objective function and the multi-user interference (MUI) in the UL transmission rate constraints, the optimization problem is non-linear and non-convex and thus is in need of a numerical efficient solution.

## 3. Numerical Efficient Solution

In order to obtain a numerical efficient solution to the problem (Equation 5), we first transform the original problem to a difference of convex/concave (DC) programming problem, and then reformulate the DC programming problem to be a convex one relying on the successive convex approximation (SCA) method [21,22]. On this basis, the numerical efficient solution is finally obtained by using an iterative routine.

We begin with the non-convex objective function and transform it to a convex one. By introducing auxiliary variables $q_{U}$ and $q_{D,k}$, and the additional constraints
(6)$$\frac{q_{U}^{2}}{\tau }\geq \sum_{k=1}^{K}p_{U,k},$$
(7)$$\frac{q_{D,k}^{2}}{\tau_{k}}\geq p_{D,k},$$
the objective function can be replaced by its upper bound of $q_{U}^{2}+\sum_{k=1}^{K}q_{D,k}^{2}$. It is easy to see that the newly defined objective function and the terms on both sides of the constraints (Equation 6) and (Equation 7) are convex [23], and thus the associated constraints are in DC form.

Now we convert to the non-convex constraint C1 and express it to be an equivalent form as
(8)$$\begin{array}{cc} \; & \log_{2}\left(\sum_{j=k}^{K}{\left|h_{j}\right|}^{2}p_{U,j}+\sigma_{\mathrm{AP}}^{2}\right)-\\ \; & \log_{2}\left(\sum_{j=k+1}^{K}{\left|h_{j}\right|}^{2}p_{U,j}+\sigma_{\mathrm{AP}}^{2}\right)\geq \frac{{\bar{R}}_{U,k}}{\tau }. \end{array}$$

In the constraint (Equation 88), the terms on the left-hand side are the differences in two concave functions and the term on the right-hand side is convex, so the constraint also owns a DC form.

The DL transmission rate C2 can be equivalently expressed as a convex one written by
(9)$$\begin{array}{c} \log_{2}\left(1+\frac{{\left|h_{k}\right|}^{2}p_{D,k}}{\sigma_{k}^{2}}\right)\geq \frac{{\bar{R}}_{D,k}}{\tau_{k}}. \end{array}$$

Now that the constraints (Equation 6)–(Equation 88) are in DC form, and the other constraints together with the newly defined objective function are convex, the original problem (Equation 5) is accordingly transformed to be a DC programming problem given by
(10)minτ,τk,pD,k,pU,k,qU,qD,kqU2+∑k=1KqD,k2s.t.6,7,8,(9),C3,C4,C5,s.t.k=1,2,⋯,K.

It is worth noting that although the objective function from problem (Equation 5) to problem (Equation 110) is newly formulated by introducing additional constraints (Equation 6) and (Equation 7), this reformulation does not alter the optimum of the original problem in that the newly introduced constraints must be active when the problem (Equation 110) attains the optimum. We prove this by contradiction. When the constraints (Equation 6) and (Equation 110) are inactive, we can decrease $q_{U}$ and $q_{D,k}$ to make them active. The $q_{U}$ and $q_{D,k}$’s decreases can further reduce the system’s power consumption. Therefore, when the problem (Equation 110) attains the optimum, the constraints (Equation 6) and (Equation 7) must be active.

Now that the problem (Equation 110) becomes a DC programming problem, SCA can be applied to transform it to be a convex one. By using the first-order Taylor approximation, the non-convex constraints (Equation 6)–(Equation 88) can be respectively transformed to be the following convex ones given by
(11)$$\begin{array}{c} \frac{2{\bar{q}}_{U}}{\bar{\tau }}q_{U}-\frac{{\bar{q}}_{U}^{2}}{{\bar{\tau }}^{2}}\tau \geq \sum_{k=1}^{K}p_{U,k}, \end{array}$$
(12)$$\begin{array}{c} \frac{2{\bar{q}}_{D,k}}{{\bar{\tau }}_{k}}q_{D,k}-\frac{{\bar{q}}_{D,k}^{2}}{{\bar{\tau }}_{k}^{2}}\tau_{k}\geq p_{D,k}, \end{array}$$
(13)$$\begin{array}{cc} \; & \log_{2}\left(\sum_{j=k}^{K}{\left|h_{j}\right|}^{2}p_{U,j}+\sigma_{\mathrm{AP}}^{2}\right)-\log_{2}\left(\sum_{j=k+1}^{K}{\left|h_{j}\right|}^{2}{\bar{p}}_{U,j}+\sigma_{\mathrm{AP}}^{2}\right)\\ \; & \;\;-\frac{\sum_{j=k+1}^{K}{\left|h_{j}\right|}^{2}\left(p_{U,j}-{\bar{p}}_{U,j}\right)}{\left(\sum_{j=k+1}^{K}{\left|h_{j}\right|}^{2}{\bar{p}}_{U,j}+\sigma_{\mathrm{AP}}^{2}\right)\ln2}\geq \frac{{\bar{R}}_{U,k}}{\tau }, \end{array}$$
where ${\bar{q}}_{U}$, ${\bar{q}}_{D,k}$, ${\bar{p}}_{U,j}$, $\bar{\tau }$, and ${\bar{\tau }}_{k}$ are the points at which the first-order Taylor’s expansions of the associated functions are carried out.

Based on these transformations, the original non-convex problem (Equation 5) can be reformulated as the following convex one: (14)minτ,τk,pD,k,pU,k,qU,qD,kqU2+∑k=1KqD,k2s.t.9,11,12,(13),C3,C4,C5,s.t.k=1,2,⋯,K.

The reformulated convex optimization problem (Equation 114) can either be solved by the interior point method or by the on-the-shelf software CVX [24]. Since the optimization problem is obtained under a group of fixed optimization variables $\left\{{\bar{q}}_{U},{\bar{q}}_{D,k},{\bar{p}}_{U,k},\bar{\tau },{\bar{\tau }}_{k}\right\}$, an iterative routine must be involved for the final optimized solution. The overall iterative procedure is given by Algorithm 1.
**Algorithm 1** SCA iterations for the problem (Equation 114).1:Initialize $\left\{{\bar{q}}_{U},{\bar{q}}_{D,k},{\bar{p}}_{U,k},\bar{\tau },{\bar{\tau }}_{k}\right\}$, set the iteration index n=0 and the maximal iteration number *N*.2:**repeat**3:   n←n+14:   solve problem (Equation 114), and obtain the optimized variables $\left\{\tau ,\tau_{k},p_{D,k},p_{U,k},q_{U},q_{D,k}\right\}$5:   update $\left\{{\bar{q}}_{U},{\bar{q}}_{D,k},{\bar{p}}_{U,k},\bar{\tau },{\bar{\tau }}_{k}\right\}$ from the solution obtained above6:**until** the algorithm converges n=N7:**return** the design variables $\left\{\tau ,\tau_{k},p_{D,k},p_{U,k}\right\}$

*Algorithm Initialization*. In Algorithm 1, the optimization problem (Equation 114) is feasible only when the fixed values $\left\{{\bar{q}}_{U},{\bar{q}}_{D,k},{\bar{p}}_{U,k},\bar{\tau },{\bar{\tau }}_{k}\right\}$ are in the feasible set of the problem. While randomly chosen $\left\{{\bar{q}}_{U},{\bar{q}}_{D,k},{\bar{p}}_{U,k},\bar{\tau },{\bar{\tau }}_{k}\right\}$ are generally outside the feasible set, an alternative initialization method becomes a must in order for Algorithm 1 to work properly. We present a simple yet efficient initialization method consisting of two steps. The first step is to pre-allocate the time resource among the UL, the DL, and the DL TDMA slots. For this, we can simply assume both equal-length two-way transmissions and equal-length DL TDMA transmissions, and in this case, we have $\tau =\frac{1}{2}$, $\tau_{k}=\frac{1}{2K}$ for any *k*. Certainly, other time allocation proportions can also be assumed as long as the time constraints C3, C4, and C5 are satisfied. Once the time resource is allocated, the joint power and time allocation problem (Equation 5) becomes a convex power allocation problem, and then the power allocation can be achieved in the associated convex set.

*Algorithm Complexity*. Now we evaluate the computational complexity of the iterative algorithm. The iterative procedure of Algorithm 1 shows that the computation complexity is mainly determined by the solving of problem (Equation 114). According to [25], the computational load for solving a convex optimization problem with *M* optimization variables is $O\left(M^{3.5}\right)$. For the considered problem (Equation 114), the set of optimization variables is given by $\left\{\tau ,\tau_{k},p_{D,k},p_{U,k},q_{U},q_{D,k}\right\}$ that consists of M=4K+2 members. After considering the iterations *N*, the total computational load of Algorithm 1 is given by $O\left(N{\left(4K+2\right)}^{3.5}\right)$. As will be exhibited in the simulations, the algorithm converged only after several round of iterations, and thus the polynomial complexity with practical *K* is very low and the proposed method is feasible.

## 4. Simulation Analysis

Since NoMA’s detection complexity and delay increase with the number of terminals, a pair of user terminals, i.e., K=2, are the most popular combination for NoMA. As a compromise, we choose K=3 in the simulations. For the network topology, we considered the same scenario as adopted by [26]. In this scenario, all the communication nodes are distributed in a circular area of a 10 m radius, where the AP is located at the origin and the three terminals are uniformly distributed over three circular annulus with the inner radii of 1, 4, and 7 m, and the outer radii of 3, 6, and 10 m, respectively. The channel gain is defined as hk2=β(dkdkd0d0)−α×RiceL, where $\mathrm{Rice}\left(L\right)$ is the small-scale channel gain with a rice factor of *L*; $d_{k}$ is the transmission distance of the IoD *k*; $d_{0}$, β, and α are the reference distance, the channel gain at the reference distance, and the path loss exponential, respectively. In the simulations, L=10, $d_{0}=1$, β=−10dB, and α=2.0; and without loss of generality, let ${\bar{R}}_{D}\overset{\Delta }{\;=}{\bar{R}}_{D,k}$, ${\bar{R}}_{U}\overset{\Delta }{\;=}{\bar{R}}_{U,k}$, $\sigma_{\mathrm{AP}}^{2}=\sigma_{k}^{2}=10^{-6}\;W$, for ∀k.

We first investigated the convergence performance of the iterative Algorithm 1. Under three transmission rate combinations $\left({\bar{R}}_{D},{\bar{R}}_{U}\right)$ with bits/s/Hz unit, Figure 2 and Figure 3 plot the numerical relation curves between the system power and the UL time allocation proportion. Although only the UL time allocation proportion is given in the simulations, the DL time allocation proportion can be readily yielded by subtracting the UL time allocation proportion from one. Simulations from these plots show that both the system power and the time allocation proportion converge at a rapid speed. Only after several iterations, the system power and the time allocation proportion remained unchanged.

Next we examined the numerical relationship between the time allocation proportion and the traffic difference on the DL and UL, as shown in Figure 3. For $\left({\bar{R}}_{D},{\bar{R}}_{U}\right)=\left(2,2\right)$, the UL time allocation proportion was about 46%, basically equal to the DL one of about 54%. For $\left({\bar{R}}_{D},{\bar{R}}_{U}\right)=\left(1,3\right)$ and $\left({\bar{R}}_{D},{\bar{R}}_{U}\right)=\left(3,1\right)$, the time allocation proportions for the UL and DL differed greatly. Specifically, for $\left({\bar{R}}_{D},{\bar{R}}_{U}\right)=\left(1,3\right)$, the UL time allocation proportion was about 71.5%, far exceeding the DL one of about 28.5%, and for $\left({\bar{R}}_{D},{\bar{R}}_{U}\right)=\left(3,1\right)$, the UL time allocation proportion was about 22%, much less than the DL one of about 78%. This is attributed to the asymmetric traffic on the DL and UL. For $\left({\bar{R}}_{D},{\bar{R}}_{U}\right)=\left(2,2\right)$, the traffic levels on the DL and UL are symmetric, and thus the DL and UL own almost the same transmission time. For $\left({\bar{R}}_{D},{\bar{R}}_{U}\right)=\left(1,3\right)$, the UL traffic is heavier than the DL, and of course requires more transmission time than the DL. On the contrary, for $\left({\bar{R}}_{D},{\bar{R}}_{U}\right)=\left(3,1\right)$, the DL traffic is heavier than the UL, and naturally requires more transmission time than the UL. The variations of the transmission time proportion with the traffic difference on the DL and UL just reflect the resource allocation of the optimization procedure.

Next we examine different resource allocation methods for the proposed system. To be specific, we investigate two methods of the joint power and time allocation and the power allocation with fixed time allocation proportion. For the power allocation with fixed time allocation proportion, the overall transmission time is firstly divided equally on the DL and UL, and then the allocated DL transmission time is further equally divided into *K* intervals to support the DL TDMA. Thus qualitatively, $\tau =\frac{1}{2}$, $\tau_{k}=\frac{1}{2K}$, ∀k. Figure 4 presents the system power performance for the two resource allocation methods under symmetric traffic levels on the DL and UL, i.e., $\bar{R}\overset{\Delta }{\;=}{\bar{R}}_{D}={\bar{R}}_{U}$. Simulations from Figure 4 show that the joint method greatly reduces the system’s power consumption as compared with the power allocation method under fixed time allocation. Obviously, the additional freedom of transmission time in the joint method allows for further reduction in the system’s power.

Finally, we compare our proposed hybrid NoMA/OMA scheme with the pure OMA scheme and the pure NoMA scheme. In the pure OMA scheme, TDMA was applied on both the DL and UL, and the joint power and time allocation for each user were still adopted for the system design, as in the hybrid NoMA/OMA scheme. In the pure NoMA scheme, NoMA was applied on both the UL and DL, and the joint power and time allocation were adopted as well. Figure 5 illustrates the system’s power consumption for the three schemes under three distinct DL and UL rate combinations. For the convenience of illustration, the hybrid NoMA/OMA scheme, the NoMA scheme, and the OMA scheme are labeled with ‘hybrid’, ‘NoMA’, and ‘OMA’, respectively, in the figure. Admittedly, when the system can afford the NoMA MUD on both the UL and DL, the hybrid scheme is inferior to the pure NoMA scheme. When the NoMA MUD cannot be affordable for the IoDs, the proposed scheme performs better than the OMA scheme. Quantitatively, for the transmission rate combinations $\left({\bar{R}}_{D},{\bar{R}}_{U}\right)=\left(1,3\right)$, $\left({\bar{R}}_{D},{\bar{R}}_{U}\right)=\left(2,2\right)$, and $\left({\bar{R}}_{D},{\bar{R}}_{U}\right)=\left(3,1\right)$, the system power of the OMA scheme is almost 3 times, 2 times, or 1.4 times that of the hybrid scheme, respectively; or equivalently, the proposed scheme saves nearly 67%, 50%, and 30% power as compared with the OMA scheme, respectively, implying that the more demanding the UL rate requirement, the greater the advantage of the hybrid scheme over the OMA scheme. Fortunately, in most low-cost IoT applications, such as wireless sensor networks, the UL traffic is dominating on the two-way transmissions, and thus a power efficient system can be reached in these IoT application scenarios. This result is not difficult to understand in that on the UL, NoMA and OMA are respectively applied in the hybrid scheme and the OMA scheme, and the superiority of the hybrid scheme over the OMA scheme is actually the superiority of NoMA over OMA. As a compromise, the hybrid scheme not only achieves NoMA gains by exploiting the rich computational resources at the AP, but also makes the IoD receivers realizable by using low-complexity OMA.

## 5. Conclusions

A two-way IoT transmission scheme combined with scalable TDD and hybrid NoMA/OMA was proposed to address the asymmetric QoS requirement and the unbalanced receivers processing capability on the UL and DL. By means of joint optimization of the transmit power and the transmit time between the UL and DL, the optimal allocation of these transmission resources was reached, thereby achieving the adaptation of transmission resources to the asymmetric QoS requirement. By hybrid NoMA/OMA, the benefits of performance improvement and complexity reduction are achieved on the UL and DL, respectively. The proposed hybrid multi-access scheme and joint resource allocation method performed better than the conventional OMA scheme with joint resource allocation.

## Figures and Tables

**Figure 1 entropy-24-01756-f001:**
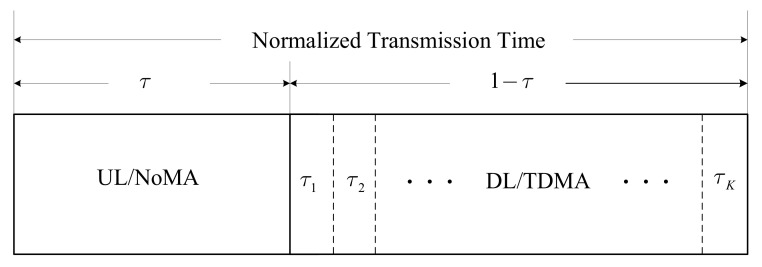
Transmission protocol for the proposed hybrid NoMA/OMA scheme.

**Figure 2 entropy-24-01756-f002:**
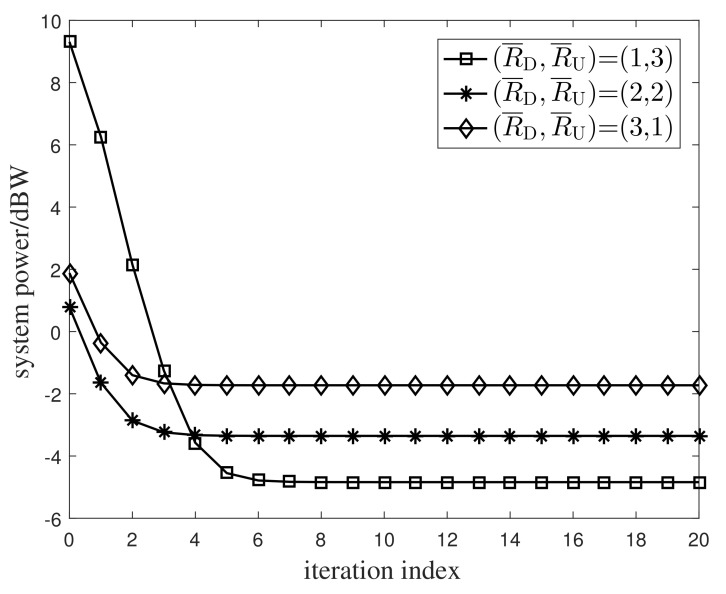
Convergence property of the system’s power.

**Figure 3 entropy-24-01756-f003:**
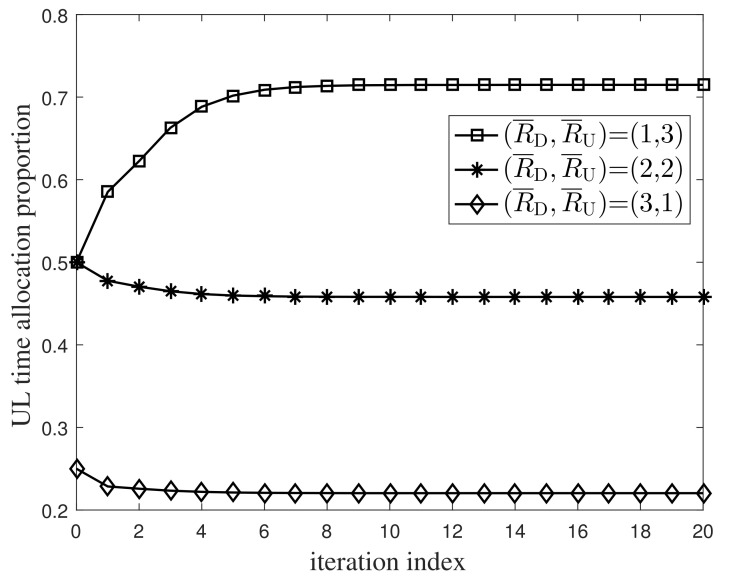
Convergence property of the uplink’s transmission time.

**Figure 4 entropy-24-01756-f004:**
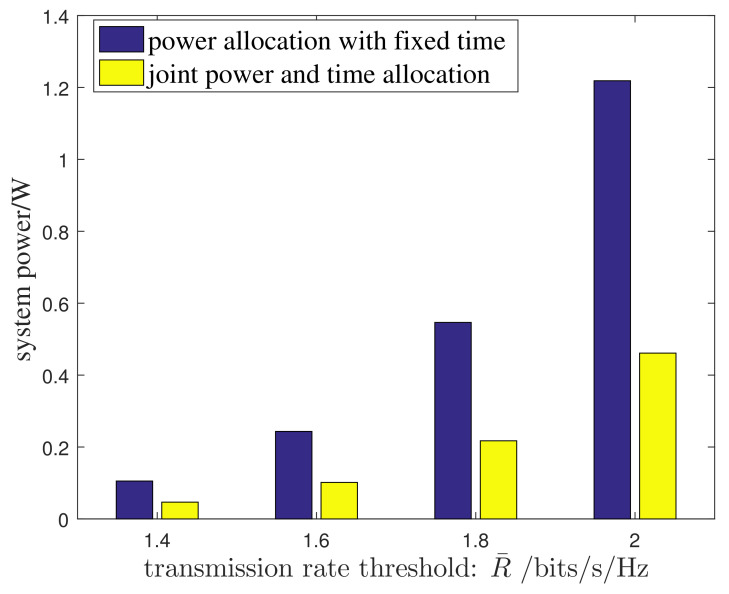
Performance comparison between the joint resource allocation and the power allocation under fixed time allocation.

**Figure 5 entropy-24-01756-f005:**
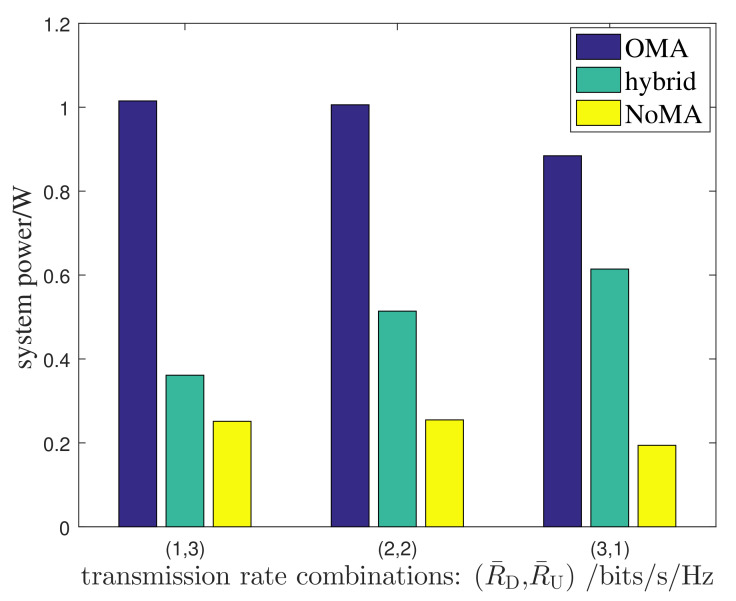
Performance comparison between the proposed hybrid NoMA/TDMA scheme and the OMA scheme.

## Data Availability

Data are contained within the article.

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
