# Peer review of "Joint Power and Time Allocation in Hybrid NoMA/OMA IoT Networks for Two-Way Communications"

_entropy, 2022, doi:10.3390/e24121756_

Round 1

Reviewer 1 Report

This paper studies joint power and time allocation for the hybrid NOMA/OMA IoT networks in two-way communications. The resulting non-convex problem is solved using the iterative successive convex approximation method. I have the following major comments.

- Though the NOMA has been studied for the past few years, does it have practical relevance for future 6G networks? Especially, when we consider the fact that MU-MIMO is being increasingly used. 

- The abstract is not very elaboratory. For instance, the contributions are unclear from the abstract. The authors should clearly describe what kind of problem is being formulated and solved and why.

-The authors considered the decoding order based o the descending channel gains. Is it always the practical choice to determine the decoding order? It should also depend on the operating rate region.

- Single antennas have been considered even at the access point. It could have multiple antennas. Alternatively, in what application scenarios the AP has a single antenna?

-The literature review should include the following relevant works on two-way communication:

[R1] Wireless Power Transfer in Two-Way AF Relaying with Maximal-Ratio Combining under Nakagami-m Fading

[R2] Hybrid NOMA/OMA-Based Dynamic Power Allocation Scheme Using Deep Reinforcement Learning in 5G Networks.

Reviewer 2 Report

In this paper, authors introduced a transmission scheme for IoT. Authors considered the scalable TDD and hybrid NoMA/OMA, as well as QoS requirement.  By combining hybrid multiple access scheme with a joint power and time allocation method, authors can solve the problem of asymmetric transmission traffics and unbalanced receiver processing. The numerical results were reported in order to confirm the correction of the method. I have some comments as follows:

+ From the practical point of view, how about the the feasibility of the proposed approach.

+ I am very interested in the power efficient of your proposed transmission. It is better to discuss further about it.

+ Authors selected K = 3 in the simulations. When the value of K increases, is there any additional difficulties/limitations?

+ English language and style are spell check required. For example, in line 26, “On this account, some work studied relay aided two-way transmissions for the NoMA based wireless networks [7–9].” should be “On this account, some works ...” etc.

Round 2

Reviewer 1 Report

The authors satisfied my concerns.

However, the response to the first comment is not very convincing. As a minor comment, the author should, in the revised manuscript, clearly lay out the relevance of NOMA in 6G/B5G networks.
